# Key components of post-diagnostic support for people with dementia and their carers: A qualitative study

Claire Bamford[1]*, Alison Wheatley[1], Greta Brunskill[1], Laura Booi[1], Louise Allan[2], Sube Banerjee[3], Karen Harrison Dening[4], Jill Manthorpe[5], Louise Robinson[1], on behalf of the PriDem study team[¶]

1 Faculty of Medical Sciences, Population Health Sciences Institute, Campus for Ageing and Vitality, Newcastle University, Newcastle upon Tyne, United Kingdom, 2 South Cloisters, University of Exeter, Exeter, United Kingdom, 3 Office of Vice Chancellor, University of Plymouth, Plymouth, United Kingdom, 4 Dementia UK, London, United Kingdom, 5 NIHR Policy Research Unit in Health and Social Care Workforce, The Policy Institute at King's, King's College London, London, United Kingdom

¶ Membership of the PriDem study team is provided in the Acknowledgments.
* claire.bamford@ncl.ac.uk

**Data Availability Statement:** All relevant data have been uploaded to a public repository: https://doi.org/10.25405/data.ncl.c.5718116.v1.

## Abstract

### Background

There has been a shift in focus of international dementia policies from improving diagnostic rates to enhancing the post-diagnostic support provided to people with dementia and their carers. There is, however, little agreement over what constitutes good post-diagnostic support. This study aimed to identify the components of post-diagnostic dementia support.

### Methods

We adopted a qualitative design using interviews, focus groups and observation to explore the perspectives of key stakeholders on the content of post-diagnostic dementia support. Purposive sampling was used to identify sites in England and Wales recognised as delivering good practice. Participants included 17 people with dementia, 31 carers, 68 service managers or funders, and 78 frontline staff. Interviews and focus groups were audio recorded and transcribed for analysis. Forty-eight sessions of observation were completed and recorded in fieldnotes. Components were identified through an inductive, thematic approach and cross-checked against national guidelines and existing frameworks; they were subsequently critically reviewed by a range of experts and our mixed stakeholder panel.

### Results

Twenty distinct components of post-diagnostic support were identified, related to five themes: timely identification and management of needs; understanding and managing dementia; emotional and psychological wellbeing; practical support; and integrating support. The first and last of these were cross-cutting themes facilitating the delivery of a unique

**Funding:** This work was primarily funded by the Alzheimer's Society, grant number 331 with small amounts of additional funding from an NIHR Senior Investigator award held by Louise Robinson, grant number NF-SI-0616-10054, and a grant from North East Commissioning Service NIHR Research Capacity Funding, grant number PO826023145.

**Competing interests:** The authors have declared that no competing interests exist.

**Abbreviations:** CST, Cognitive Stimulation Therapy; GP, General Practitioner; NICE, National Institute for Health and Care Excellence.

constellation of components of post-diagnostic support to each individual living with dementia or dyad at a particular time.

## Conclusions

Our work offers an empirically based framework to inform the development and delivery of holistic, integrated and continuous dementia care from diagnosis to end of life. It highlights the relevance of many components to both people living with dementia and their carers. Since the framework was developed in England and Wales, further research is needed to explore the relevance of our components to other sectors, countries and care systems.

## Introduction

Improving post-diagnostic dementia care and support requires a shared understanding of what this comprises and how it can best be delivered. Although post-diagnostic support has been defined as *a 'system of holistic, integrated continuing care in the context of declining function and increasing needs of family carers'* [1], such definitions have limited value in operationalising and delivering support in practice. Similarly, although an increasing number of countries and the World Health Organisation have developed national strategies or guidelines on dementia these often provide little practical detail to inform service development [2–4]. Recent work has highlighted a lack of shared understanding over the meaning and content of post-diagnostic support in England and Wales; while some professionals use a similar definition to that cited above, others conceptualise post-diagnostic support as one or two meetings immediately after diagnosis [5]. The need for greater clarity over the components of care required to support people with dementia was identified as the top priority for research by the James Lind Alliance Dementia Priority Setting Partnership [6].

Finding affordable and sustainable ways to deliver post-diagnostic support to enable people and their carers to live as well as possible with dementia is a key challenge globally [1]. Currently there is evidence of a lack of services, especially as the disease progresses, and inequities in access to post-diagnostic support [5,7–10]. In England and Wales, despite a national Dementia Well Pathway covering preventing well, diagnosing well, supporting well, living well and dying well [11], service provision remains focused mainly on the point of diagnosis and the first year afterwards. Although there is financial reimbursement for a primary care-based annual review [12], there is little guidance as to its content. Provision of post-diagnostic support has been negatively impacted by the COVID-19 pandemic [13–15], leading to reduced access to services for many people with dementia.

To meet the diverse needs of people with dementia and enable them to live as well as possible requires supportive government policies and coordinated input from health and social care services and the voluntary and community sector, together with initiatives such as dementia friendly communities [16,17]. There are, however, barriers to the delivery of integrated care including lack of funding, fragmented and fragile services, lack of shared information systems and challenges to interprofessional relationships [18,19]. The 2016 World Alzheimer Report recommended a *task-shifted, task-shared approach* with the development, and expansion, of primary and community-based post-diagnostic dementia support services, enabling secondary care to lead on diagnostic assessment and complex case management [1]. A key precursor to deciding which aspects of care are appropriate for task-shifting or task-sharing is clarity over the nature of such tasks and roles [20,21].

As part of a wider research programme we conducted extensive qualitative research (interviews, focus groups and observation) to identify the components of post-diagnostic support in dementia, highlight examples of good practice, and identify factors influencing implementation of each component.

## Methods

We have followed the Standards for Reporting Qualitative Research [22] and Consolidated Criteria for Reporting Qualitative Research [23] in describing the methods (see S1 File).

### Sample

The study had two phases. In Phase 1, we focused on the views of professionals responsible for developing, commissioning (funding), and managing services for people with dementia in England and Wales. We used internet and literature searches to identify sites recognised as providing 'good practice', operationalised as award-winning services or those cited in policy documents. Additional sites were identified through the recommendations of participants and an e-survey of commissioners [24].

In Phase 2, we selected six of these services for in-depth case study, focusing on sites with clear links to primary care and which reflected the diversity of current provision of post-diagnostic support in terms of providers (primary care, secondary mental health care and third sector), components and links with other sectors and services. Further details of methods are reported elsewhere [5,19].

### Recruitment

The study received ethical approval from National Health Service Research Ethics Committee Wales 3 (reference 18/WA/0349). The initial approach to potential Phase 1 participants was made by email, excluding those who did not respond after three contacts. Initial discussions regarding Phase 2 participation were made via email with Phase 1 participants. Approvals from all sites were in place prior to data collection. Participants were not previously known to the researchers. Recruitment information included name, role and, in Phase 2, photographs of the researchers. People with dementia and carers were initially approached by gatekeepers, and opted into the study by mail. Due to research governance delays, data collection was compressed into a limited time period; while we included all available participants, we cannot be certain that we achieved data saturation.

### Data collection

Topic guides for interviews and focus groups were developed and piloted with our mixed stakeholder panel of people with dementia, carers and professionals convened to inform our research and ensure that it was situated in real life experiences. The areas of post-diagnostic support included in the service user topic guide were also developed with this group. In Phase 1 we explored current services provided or commissioned; perceived gaps in services; views on interventions recommended in the National Institute for Health and Care Excellence (NICE) dementia guidelines; and views on the role of primary care in post-diagnostic support. Phase 2 interviews explored views on the six selected services in detail, including links with primary care, social care and other providers and the range of interventions provided. Further details are provided in S2 File. Data were collected between January and December 2019 (CB, interviews with people with dementia and carers; GB, interviews, focus groups and observation and AW, interviews, focus groups and observation). Phase 1 interviews were conducted by

telephone, with one face-to-face focus group. The majority of Phase 2 data were collected face-to-face in a place of participants' choice, usually home (for service users) or place of work (for professionals), with a small number of interviews conducted by telephone. Cohabitees were occasionally present during interviews with service users. Three Phase 1 participants also took part in a follow-up focus group or interview in Phase 2. Observation included direct service provision and relevant meetings.

Written consent was obtained from most participants in telephone interviews (with verbal consent recorded where the consent form was not returned prior to the interview). All participants in interviews and focus groups gave written consent; prior to observation professionals gave written consent while people with dementia and carers gave verbal consent. No identifiable information was recorded about people with dementia and carers during observation. Interviews and focus groups were audio recorded, transcribed (by an external company; all data were transmitted over secure connections), checked and pseudonymised for analysis (CB, GB, AW). Field notes of observation sessions were made as close to contemporaneously as possible, then written up and pseudonymised for analysis (GB, AW). Data were not returned to participants for checking. Focus groups lasted on average 48 minutes (Phase 1) and 54 minutes (Phase 2); the average duration of interviews was 35 minutes (Phase 1) and 37 minutes (Phase 2).

## Analysis

We conceptualised components as any description of *what* was delivered to people with dementia and carers as a part of post-diagnostic support. We deliberately did not focus on *who* delivered support or services since similar components may be delivered by professionals with a range of skills and backgrounds, and we did not wish to impose constraints on how components were delivered or by whom. We used an inductive, thematic approach to analysis, developing a coding frame iteratively through a series of workshop discussions (CB, LB, GB, and AW) prior to systematically coding the data (GB, AW, Phase 1; LB, AW, Phase 2) in Nvivo. We additionally identified components through systematic searching of the data. Once we had retrieved all of the components we sorted them into a single list, combining them and removing duplication as required. We then crosschecked the components against guidelines and frameworks [11,25,26], as well as related interventions such as end-of-life care [27] and Admiral nursing (specialist dementia nursing [28]), and reviews conducted as part of the PriDem study [29–31]. Our intention in doing this was not to add components that were not represented in our data, but rather to explore the ways in which other authors had conceptualised post-diagnostic dementia care and support to facilitate comparison with previous work.

**Reflexivity.** All team members were female health services researchers, with diverse backgrounds in psychology, social gerontology and sociology as well as personal experience of dementia caregiving. Three of the team (GB, LB, AW) were early career researchers (2–5 years' experience); CB had 25 years' experience of dementia care research. This experience was valuable in the design of the study, recruitment of participants and data collection. During analysis we used an inductive approach which was grounded in the data, to minimise the impact of the researchers on the findings. One team member involved in analysis was not involved in data collection (LB); this allowed for a fresh perspective which was not coloured by pre-conceptions from immersion in the field. To enhance trustworthiness, the components were critically reviewed by the wider study team, representatives of the Alzheimer's Society and our mixed stakeholder panel. Each of these three groups was asked to check the list for coherence and identify any overlaps or duplication. At each stage, the list was refined.

# Results

## Participants

**Professionals.**    In Phase 1 we recruited 61 service managers and commissioners (Table 1) [5]. An additional 36 potential participants were contacted but not recruited, most commonly due to non-contact (n = 23); other reasons were that they were not suitable/had left the post (n = 6) or were unavailable within the timeframe (n = 7). In Phase 2 all senior staff/managers approached for interview took part (n = 7). Ten frontline staff were interviewed and 42 front-line staff took part in five focus groups (of 5 to 10 participants). Since we relied on service managers to distribute study information on our behalf, we are unable to quantify non-response. Finally, we approached 41 professionals who liaised or worked with the service but were employed by other organisations, including general practitioners (GPs), social prescribers (employed in primary care to link patients to community resources), community matrons and care home staff. Twenty six linked professionals were recruited (4 did not return the opt-in form, 10 did not respond and 1 was unavailable). We undertook 48 sessions of observation with 39 professionals including initial assessments and reviews, clinic sessions, and group interventions with service users, and internal and multidisciplinary team meetings. A total of 84 professionals took part in Phase 2, some of whom contributed different types of data. Details of sector and role are shown in Table 1; cross sector staff worked across secondary care and third sector organisations; primary and secondary care; or health and social care.

**Service users.** In Phase 2 we approached 118 service users (61 people with dementia of whom 17 were interviewed and 57 carers of whom 31 were interviewed). Reasons for non-recruitment were that potential participants: did not return the opt-in form or returned it too late (35 people with dementia and 20 carers); did not respond to telephone calls to arrange an interview (2 people with dementia and 5 carers); were unable to give informed consent (6 people with dementia); or were unavailable within the time frame (1 person with dementia and 1 carer).

**Table 1.  Characteristics of participating professionals.**

| | | | Phase 1 (n = 61) | Phase 2 (n = 84) |
|---|---|---|---|---|
| Sector: | | Primary care | 20 | 15 |
| | | Secondary care | 12 | 41 |
| | | Community health | 2 | 0 |
| | | Social care | 9 | 3 |
| | | Third sector | 11 | 4 |
| | | Cross sector | 7 | 20 |
| | | Private sector | 0 | 1 |
| Role: | | Commissioners/service development leads | 25 | 0 |
| | | Service managers | 25 | 7 |
| | | Old age psychiatrist | 1 | 1 |
| | | GP | 0 | 8 |
| | | Specialist nurse | 6 | 23 |
| | | Non-specialist nurse | 0 | 1 |
| | | Allied health professional | 2 | 11 |
| | | Dementia navigator | 2 | 26 |
| | | Social worker | 0 | 5 |
| | | Non-specialist support workers | 0 | 2 |

**Table 2. Characteristics of participating people with dementia and carers.**

| | | People with dementia (n = 17) | Carers (n = 31) |
|---|---|---|---|
| Gender: | Male | 10 | 8 |
| | Female | 7 | 23 |
| Age (years): | 40 <50 | 0 | 4 |
| | 50 < 60 | 0 | 6 |
| | 60 < 70 | 2 | 5 |
| | 70 < 80 | 7 | 9 |
| | 80 < 90 | 6 | 4 |
| | 90+ | 2 | 0 |
| | Not disclosed | 0 | 3 |
| Dementia subtype: | Alzheimer's disease | 6 | 16 |
| | Vascular dementia | 1 | 0 |
| | Frontotemporal dementia | 0 | 3 |
| | Lewy body dementia | 2 | 3 |
| | Young onset dementia (unknown type) | 0 | 1 |
| | Young onset dementia (Frontotemporal) | 2 | 2 |
| | Mixed dementia | 0 | 4 |
| | Mild cognitive impairment | 3 | 0 |
| | Unknown by participant | 2 | 2 |
| | Not disclosed | 1 | 0 |
| Time since diagnosis: | < 1 year | 3 | 3 |
| | 1 < 2 years | 5 | 8 |
| | 2 < 5 years | 5 | 10 |
| | 5+ years | 2 | 10 |
| | Not disclosed | 2 | 0 |
| Living arrangements (people with dementia): | Alone | 9 | n/a |
| | With spouse or family | 8 | n/a |
| Co-resident with person with dementia (carers): | Yes | n/a | 15 |
| | No | n/a | 16 |

Table 2 illustrates the breadth of the sample of people with dementia and carers in terms of gender, age, type of dementia, time since diagnosis and living arrangements. The age of participating people with dementia ranged from 66 to 96 (mean 80 years), with diagnosis having been made on average just over two years prior to the interview. The age of carers ranged from 42–87 (mean 65 years), on average the person they supported had been diagnosed for just over three years.

Case study sites varied in terms of provider (National Health Service, mental health Trust, primary care, third sector) and scope [19]. A brief description is given below:

- Ongoing review and support by an Admiral Nurse (a specialist dementia nurse) based in a single general practice

- Ongoing review for all people with dementia registered with local general practices by GPs with specialist dementia training

- Secondary NHS care led models linked to specific GP practices (two sites), providing ongoing review, interventions and easy access to specialist staff (e.g. nurse, occupational therapist, clinical psychologist, consultant old age psychiatrist) as and when needed

- Secondary care led enhanced memory assessment service offering group interventions and some ongoing support via a clinical drop-in service
- Third sector community-based non-clinical dementia navigators providing information and linking people with dementia and carers to local services and groups tailored to their interests.

## Key themes and components of post-diagnostic support

Two crosscutting themes–'timely identification and management of current and future needs' and 'integrating support'–were identified. These facilitate the delivery of the unique constellation of components of post-diagnostic support relevant to each individual person with dementia or dyad at a particular time. The remaining components were grouped into three themes (Fig 1). Each theme is described in more detail below. While carer support was included as a separate theme in early versions of the analysis, it became clear that many components within each of the five main themes were relevant to both people with dementia and carers. Rather than having a separate theme relating to carers, we have incorporated components for people with dementia, dyads and carers into each of the themes. This is indicated in by the use of colour in Fig 1. Where components are identified as relevant to both people with dementia and carers, this does not mean that their needs are identical; they may, for example, have very different needs and preferences regarding information provision. Definitions of each component and examples of good practice are provided in Tables 3–7; we have also used asterisks within these tables to denote practice consistent with NICE dementia guidelines [25].

**Timely identification and management of current & future needs.** The illness trajectory of dementia varies across and within subtypes; this heterogeneity means that tailored support is essential from diagnosis to death. This requires regular review to identify emerging needs and plan how to address them. Components within this theme are described in Table 3; the text below highlights key issues relating to the practical delivery of the components.

While most of our case study sites provided six-monthly reviews, observation highlighted variability between practitioners in the content and depth of review, suggesting that explicit attention is needed to ensure a consistent approach. Planning for contingencies and future needs often received limited attention; useful ways to approach such discussions included planning for enjoyable events or thinking through practical solutions in the event of carer unavailability. The importance of regular review was highlighted by the reluctance of some people with dementia and carers to seek help. In some cases this reflected concerns over unnecessarily bothering professionals; in others, people with dementia and carers were unsure whether symptoms were related to dementia, did not recognise their own needs, or were unaware that help was potentially available:

> I think the trouble is, at the beginning, you're not in any fit place to be able to decide, "Yes, I need help," or whatever. "I'm going to go and see somebody." This is what I said about a therapist or call it what you like–I don't care–to help you cope with it, and, of course, [my wife] for her as well, because, as I say, it did–it just turned our lives upside down. (P202, person with dementia)

A range of issues relating to the practical organisation of all components within this theme was identified. Having sufficient time, without feeling rushed, was key to exploring and addressing complex problems and support needs, and ensured that people with dementia and carers did not feel pressurised:

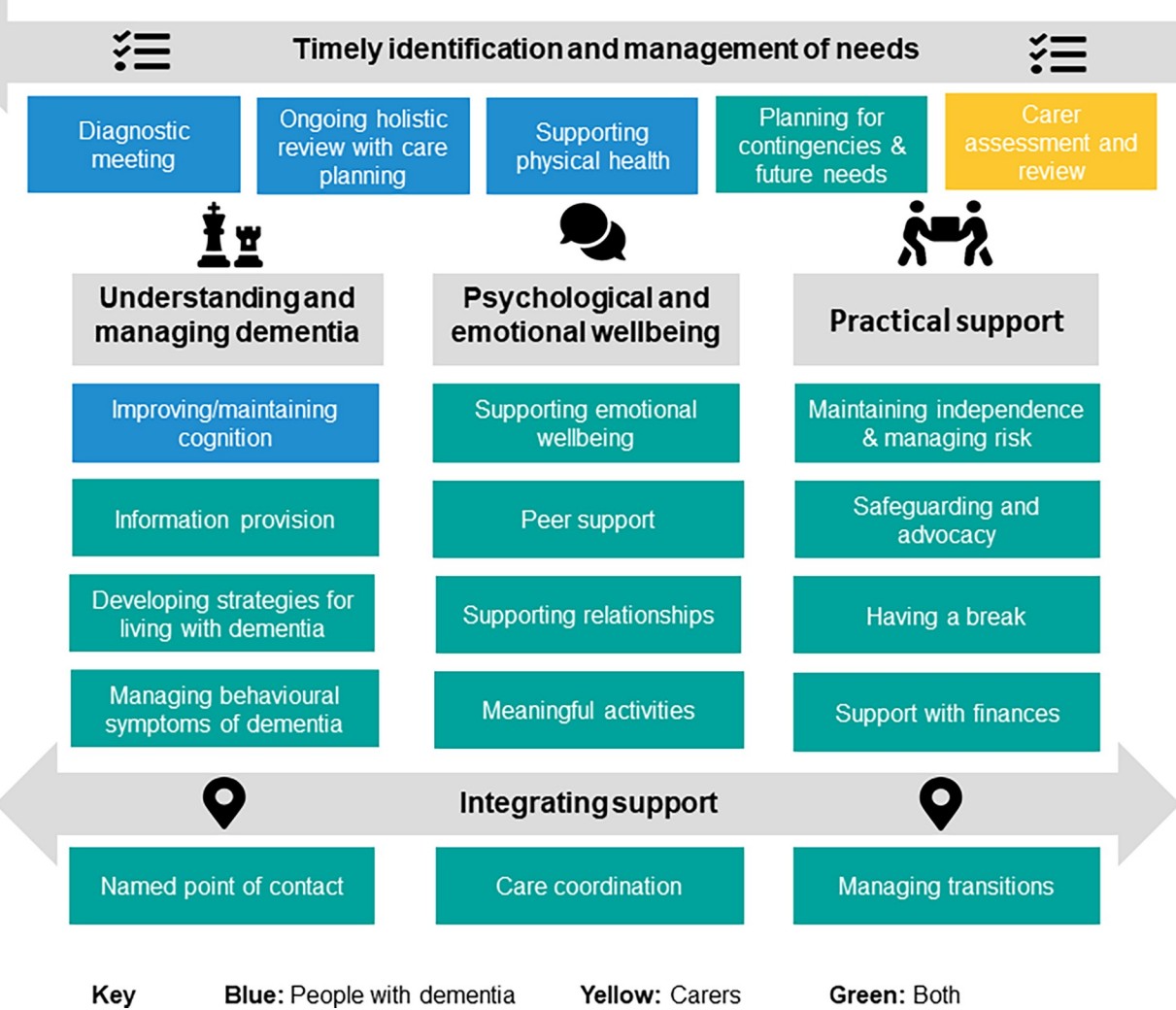

**Fig 1. Themes and components of post-diagnostic dementia support.**

*I never felt with [Admiral Nurse] that we were taking an appointment. . . she had more time for us as well, that's another thing, because GPs have allocated times, haven't they, for seeing patients. Where time never felt the same issue when we went to see [Admiral Nurse] or she came to see Mum and Dad. You never felt, "Right, you've had your five minutes. Right, out we go." (C404, carer)*

Flexibility over the place of appointments was valued, as were opportunities for people with dementia and carers to be seen separately. Most participants agreed that the frequency of review should be tailored, with more frequent reviews as more complex needs emerged. Some staff, however, worried that this would create dependency. Some services offered informal support between reviews by visiting existing groups in community locations. This low-key approach facilitated early identification of changes:

*[. . .] other patients often pick up if someone has not been there or if they've seen any change in someone, they quite often say, "Oh, so and so is not quite their normal self," or "we've seen*

**Table 3. Components relating to timely identification and management of needs.**

| Component | Description | Examples of good practice |
|---|---|---|
| Diagnostic meeting | One or more meetings with the service diagnosing dementia, to communicate the diagnosis, discuss treatment options, provide information and signpost. | • Negotiating the most supportive environment for the diagnostic meeting, including location and companions<br>• Checking understanding of the diagnosis and what it means to the person with dementia and carer<br>• Planning follow up visits to allow time to process the initial diagnosis and formulate questions<br>• Acknowledging that it is 'normal' to be distressed following a diagnosis of dementia, and arranging follow up to check on the adjustment process<br>• Providing information on available pharmacological and non-drug treatments*<br>• Ensuring the follow-up meeting is holistic, rather than focusing solely on medication titration<br>• Directing people with dementia and carers to relevant services for information and support (see Information Provision component for details)* |
| Holistic ongoing review with care planning | Proactive review involving a range of professionals, at flexible intervals depending on the person's needs. | • Taking a holistic approach tailored to person with dementia/carer priorities<br>• Opportunistic review to assess emerging needs*<br>• Creating a personalised care plan, which provides a record of decisions made and actions to be taken<br>• Robust arrangements for reviewing and sharing the care plan<br>• Ensuring that people with dementia and carers have an opportunity to speak individually in private<br>• Explicit plans for future contact* |
| Physical health and medication review | Ensuring that physical health problems (whether multiple conditions or those related to dementia) are managed promptly and avoiding diagnostic overshadowing. | • Medication review, including adherence and reducing cholinergic burden*<br>• Holistic approach including a focus on a healthy lifestyle, sensory impairment, foot care etc.<br>• Integration of dementia review with reviews for other long-term conditions to minimise duplication<br>• Involving allied health professionals for support with physical symptoms such as speech, and swallowing |
| Planning for contingencies and future needs | Having opportunities and support to think about goals for the future, as well as future care preferences and emergency situations. | • Setting future goals e.g. holidays<br>• Planning for changes in routine e.g. Christmas<br>• Planning for periods where a carer is unavailable e.g. hospital stay<br>• Introducing Advance Care Planning, exploring whether and how people with dementia and carers want to take this forward, and revisiting as appropriate*<br>• Providing information and support with setting up Lasting Power of Attorney* (proxy decision making)<br>• Providing access to Carers' Emergency Cards or similar schemes |
| Carer assessment and review | Formal assessment of carer needs which may lead to interventions such as breaks*, psychoeducation*, or referral to psychological or other support services. | • Easy access to formal carer assessment*<br>• Exploring carer needs as part of routine practice<br>• Ensuring carer has a chance to speak in private<br>• Proactive follow-up if circumstances change |

* Indicates example is recommended in England and Wales by NICE [25].

*a bit of a change in them". Or they can just come up and chat to me over a coffee and say, "My father is not that well," or something. I say, "Well come over and have a chat. We'll book an appointment". (S031, dementia practitioner)*

**Understanding and managing dementia.** This theme comprises components to ensure people with dementia and their carers are fully informed about their condition, what to expect, and how to live as well as possible. This includes pharmacological and non-pharmacological

**Table 4. Components relating to understanding and managing dementia.**

| Component | Description | Examples of good practice |
|---|---|---|
| Improving/maintaining cognition | Interventions to improve memory and thinking may include both pharmacological and non-pharmacological interventions. The main drug therapies were cholinesterase inhibitors and memantine. Non-pharmacological interventions include cognitive stimulation therapy (CST). | • Offering tailored CST*<br>• Teaching carers the principles of CST when people with dementia were unable to attend groups<br>• Offering pharmacological treatments for dementia as appropriate*<br>• Explaining purpose of medications to people with dementia and carers<br>• Exploring any barriers or concerns about taking medications<br>• Using prompts to identify side effects after starting medication<br>• Ensuring locations are accessible and timing is appropriate |
| Information provision | The provision of tailored, accessible information about dementia to people with dementia and carers. Information may be provided in verbal or written form or at group sessions. It includes, but is not limited to, information related to medication, driving and participating in research. | • Providing accessible information relevant to the circumstances and stage of dementia*<br>• Exploring the information preferences of people with dementia and carers, while recognising that these may be very different or conflicting<br>• Tailoring timing, content, format of information given<br>• Reviewing information needs and preferences at regular intervals<br>• Offering groups for people with specific subtypes of dementia<br>• Providing tailored information about dementia subtype, prognosis and what to expect*<br>• Providing information on professionals involved in their care and how to contact them*<br>• Providing information on driving, work, local support services and opportunities to participate in research*<br>• Ensuring information is accurate and up to date<br>• Ensuring that people with dementia/carers know how to access further information when needed, including online and national resources* |
| Developing skills and strategies for living with dementia | Interventions or activities to help people with dementia and carers to understand and manage cognitive and functional decline in dementia. These may be group or individual. Courses may be for people with dementia, carers or dyads. | • Covering coping strategies, information and signposting<br>• Input from a variety of professionals (e.g. speech and language therapy, Occupational Therapy (OT)*, psychology, dietetics)<br>• Flexible options for attending alone or with a carer if preferred<br>• Accessible locations and appropriate timing<br>• Flexible approach e.g. offering courses by telephone as well as face to face<br>• Opportunities to learn from other people with dementia and carers<br>• Offer carers of people living with dementia a psychoeducation and skills training intervention*, e.g. [32,33] |
| Managing non-cognitive symptoms of dementia | Interventions and activities to understand the antecedents and impacts of non-cognitive symptoms and to explore creative management strategies | • Using approaches such as observation or charting to identify antecedents to non-cognitive symptoms*<br>• Pooling knowledge and expertise relating to both the potential causes and management of non-cognitive symptoms through multidisciplinary team meetings<br>• Ruling out clinical causes of non-cognitive symptoms (e.g. pain, delirium)*<br>• Exploring non-pharmacological interventions and activities to manage non-cognitive symptoms*<br>• Considering pharmacological options only if alternative options have not been successful* |

* Indicates example is recommended in England and Wales by NICE [25].

treatment options, many of which are evidence-based and recommended in NICE guidelines [25]. An overview of each component is provided in Table 4.

To ensure optimal treatment, it was important to explore and address barriers and concerns about treatment options. People with dementia and carers, for example, often felt unsure of the purpose of medications they were offered and what effects they might have:

**Table 5. Components relating to psychological and emotional wellbeing.**

| Component | Description | Examples of good practice |
|---|---|---|
| Supporting emotional wellbeing | Interventions to enhance mood, support adjustment to dementia diagnosis, and manage anxiety. | • Access to cognitive behavioural therapy<br>• Access to (specialist) counselling<br>• Access to pharmacological treatment for severe anxiety and depression if required*<br>• Group work focusing on adjustment to diagnosis |
| Peer support | Opportunities to meet virtually or face-to-face with peers to share experiences, information, advice and social activities. | • Opportunities for online peer support<br>• Facilitating peer support for specific groups (e.g. dementia subtype) by expanding geographical boundaries to ensure viability |
| Meaningful activities | Access to activities/groups/clubs to ensure that people with dementia have opportunities to socialise and maintain their identity through pursuing existing hobbies and interests. It is also relevant to carers who can become isolated, particularly as dementia progresses. | • Flexibility to attend existing groups or have personalised one-to-one support with interests or activities<br>• Ensuring that generic groups are dementia-friendly<br>• Supporting engagement e.g. transport, accompanying people with dementia to activities where needed*<br>• Tailoring activities to the individual*<br>• Life story work to discover interests and engage the person |
| Supporting relationships | Recognition of the impact of dementia on couples and families, and providing interventions when needed. | • Access to (specialist) couples/family counselling, including interventions such as Living Together With Dementia [34]<br>• Flexibility to see people in different configurations according to need<br>• Sensitivity to relationship dynamics |

* Indicates example is recommended in England and Wales by NICE [25].

*They offered, at that time, medication treatment. We declined on the basis that, at this point, she was probably not severely affected and it might. . . I was a bit concerned because her other problem is continence. Some of the side effects, I was reading, of the medication might potentially have affected the continence. [. . .] It would've been useful to have a bit more discussion over how well it works, if it's likely to work. (C101, carer)*

This highlights the key role of information to support decision-making. Tailoring information to individual needs and preferences seemed an underdeveloped area of practice. We observed few attempts to explore information preferences (whether in terms of amount, level of detail or format) or to recognise that the preferences and needs of people with dementia and carers may be different. Instead, there was a tendency to rely on a one-size-fits-all approach:

*You get everything at once, in a big bundle of stuff and then you are kind of left to get on with it. (C001, carer)*

Information provision seemed to focus on the immediate post-diagnostic period; yet it was clear that many carers would have welcomed information on prognosis at a later stage in the illness trajectory. Ensuring that needs are met requires revisiting information needs or available interventions at several points in the illness trajectory; for example, carers may not be ready for a psychoeducation course immediately following diagnosis but may find it useful after having more relevant experience. Uptake of interventions such as Cognitive Stimulation Therapy (CST), was also facilitated where groups could be tailored to specific populations such as those with more severe dementia or from minority ethnic groups:

*We had a really nice group, it was a year ago, of all Caribbean ladies. So, we were able to adapt that session completely. Like, the food session, for example, everyone wanted to bring in food from the Caribbean and share it with us, which was really nice. And then, like, the*

**Table 6. Components relating to practical support.**

| Component | Description | Examples of good practice |
|---|---|---|
| Maintaining independence and managing risk | Supporting people with dementia to keep their independence with an acceptable level of risk. This includes psychological aspects of feeling independent as well as functional aspects such as mobility and activities of daily living. | • Supporting people with dementia to do as much as they are willing and able to do for themselves<br>• Educating carers and non-specialists about risk<br>• Providing support workers to help with transport, work or volunteering<br>• Arranging home care services to assist with day-to-day tasks<br>• Making sure the physical environment is suitable<br>• Providing assistive technology e.g. call alarms, tilting kettles<br>• Access to disability support e.g. RADAR keys (for accessible toilets), Blue Badge (permits to park in reserved places for people with disabilities)<br>• Supporting safety in the community, e.g. the Herbert Protocol (enabling information to be shared if a person is missing) or Dementia Guardian Angels (portable location devices) |
| Advocacy and safeguarding | Ensuring that people with dementia are involved in decisions as much as possible and that carers are supported when making difficult decisions. Ensuring that both people with dementia and carers are protected from abuse and exploitation. | • Ensuring that people with dementia are included in decision-making and that everyone's perspective is considered<br>• Involving advocacy services as and when needed<br>• Supporting carers to make decisions<br>• Seeking multidisciplinary team input on complex needs<br>• Referring to specialist services if there are concerns that the situation cannot be managed safely |
| Having a break | Opportunities for people with dementia to have a break from routine and for carers to have time off from caring. This can include supported breaks where couples are able to have a holiday together but caring responsibilities are shared or taken on by support workers. | • Flexibility in terms of length of break<br>• Access to planned and emergency breaks*<br>• Options for in-home and out-of-home respite<br>• Ensuring culturally appropriate options are available |
| Support with financial benefits and entitlements | Ensuring that people with dementia and carers receive all the benefits and financial support to which they are entitled. | • Practical help with form filling if needed, not just information provision<br>• Referral to specialist support e.g. third sector services<br>• Support with administrating funds provided for care |

* Indicates example is recommended in England and Wales by NICE [25].

*famous faces activity, you adjust it to people that they would know of. Rather than maybe, you know, English celebrities or politicians that they might not be as familiar with. So, kind of, change it as much as you can to suit the client group, whilst still keeping all of the principles. (S104, assistant psychologist co-ordinating CST)*

Providing 'living well' groups aimed at people with mild cognitive impairment or early dementia who may not yet benefit from CST was a further way of improving access to support and ensuring that coping strategies are embedded at an early stage in disease progression.

There was crossover and reinforcement between the components in this theme; for example, information and support groups were valued by one person with dementia for strengthening and supplementing self-management techniques and other coping strategies:

*He said he had previously used self-management techniques with his dementia but had got to the point where he felt like he wasn't always "winning" and needed some extra help. He said the groups had helped him to pick up on things that helped him "get the upper hand" in the "game of chess" he was playing against his dementia; he explained that knowing that certain things were symptoms of his dementia allowed him understand what was going on better and why he could notice changes in himself. (Fieldnotes of vascular dementia information group, site 6)*

**Table 7. Components relating to integrating care.**

| Component | Description | Examples of good practice |
|---|---|---|
| Named point of contact | A named health or social care practitioner* (or other single point of contact, e.g. a telephone hub) that service users and carers can contact for help and support as needed. | • Robust system for allocating & reviewing named point of contact<br>• Ensuring that the named point of contact is knowledgeable about local services<br>• Ensuring that the named point of contact is able to access and share information with other services, including care and support plans*<br>• Ensuring that the named point of contact is accessible (not necessarily an individual, contactable in various ways) and responsive |
| Care coordinator | An individual responsible for more in-depth case management of a person with dementia and liaison with other services, for example, arranging and attending 'best interests' meetings if there are safeguarding concerns or similar. | • Providing additional expertise when needed<br>• Liaising with and coordinating multiple services*<br>• More intensive oversight for a time-limited period; care stepped down to named point of contact when appropriate |
| Managing transitions | Ensuring continuity of post-diagnostic support and smoothing transition from one service to another. | • Ensuring a smooth transition between diagnostic and community services<br>• Timely and rapid access to intensive support when needed, e.g. through linking with specialist teams<br>• Access to patient information to support care in different settings*<br>• Identifying other options on completion of time-limited interventions<br>• Supporting people with dementia and carers through transitions, including hospital discharge planning and moves to residential care |

* Indicates example is recommended in England and Wales by NICE [25].

This component also encompasses the management of non-cognitive symptoms in dementia. Fieldwork indicated the value of team and multidisciplinary team meetings for sharing information, pooling knowledge of the context and generating management strategies:

*The patient was regularly shouting at night, which was causing problems with the neighbours and had put them at risk of losing their home. The nurse prescriber confirmed that she had sent a list of medications to S110 that can cause nightmares, for her to check the patient's medication against. [. . .] S108 suggested other night time care options that had worked well for another patient of hers. (Fieldnotes of team meeting, site 1)*

**Psychological and emotional wellbeing.** In addition to providing interventions for managing psychological symptoms and supporting adjustment to a dementia diagnosis, components within this theme relate to maintaining a sense of identity and purpose through peer support groups, social and leisure activities, and supporting relationships. These are described in Table 5.

Access to psychological interventions for depression and anxiety was usually via specialist mental health psychological services. In addition, peer support was valued by several people with dementia and carers since it often validated their own experiences and offered strategies. It could also lead to enduring supportive relationships:

*That has still been the most important and beneficial thing, not alone because of the information and the guidance of where you should go, but it cemented friendships with people that*

*were in the same boat. As I said, nine years on, we're all still friends and we support each other as best we can [. . .] It was the first time that I thought, "Someone knows." I think that's one of the things that- it is, somehow or other, having someone there that isn't a professional, that knows exactly what you're going through. (C302, carer)*

Two spouses of people living with young onset dementia identified peer support as a potentially valuable intervention that was not currently available to their partners. This lack of provision may reflect the difficulties of creating groups for less common subtypes of dementia; one site had addressed this problem by extending the catchment area of a support group for people with Lewy body dementia to create a group of viable size. Attending to the interests, preferences and abilities of each individual with dementia (and their carer) was essential to support engagement in appropriate meaningful activities and required a detailed knowledge of local services:

*[A dementia navigator had] read through the case notes before going in, and she'd noticed the service user liked singing and liked music. She went in with just a whole raft of all the opportunities of singing and clubs and stuff in that area. She knew that he was Irish. She went with all these Irish services. So it is like that personal care because you can tailor it to whoever you want because she knows what there is, whereas for me or one of you guys, I don't know if you know every single Irish community service there is for dementia. (S505, memory nurse)*

In addition to providing information about possible options, attention was also needed to the 'art of engagement'. Direct experience of groups or activities enabled workers to bring options to life in a way that helped people with dementia to gain a deeper understanding of what was on offer. For people with dementia who were not interested in group activities, options seemed more constrained. While described relatively infrequently, options such as befriending services or personal support workers, could enable people with dementia to continue to pursue interests or outings:

*We have a young volunteer that's been coming once a week [. . .] she either sits with my mum for an hour, an hour and a half, or takes her along the road to a little café. And they have a little bit of lunch, and then come back in again. (C003, carer)*

Several gaps in services were noted. These included services which provided opportunities for people with dementia and carers to do activities together, as well as separately; specialist services for people with young onset dementia, since their priorities were often very different, with support with employment, parenting or volunteering being valued; and provision related to minority ethnic groups and other marginalised populations:

*You've got entire communities out there who just do not have anything that's geared culturally towards what they would enjoy and would enhance their wellbeing. It's such a noticeable gap in a city like this. (S203, dementia navigator)*

In order to support relationships, professionals needed to be aware of possible tensions that could arise in relationships due to changing roles or frustration with symptoms of dementia and provide opportunities for people with dementia and carers to speak frankly about any difficulties in private. While carers often created opportunities to talk privately, either explicitly negotiating a separate conversation with staff, or by raising issues on the doorstep as the worker was leaving, we did not observe any people with dementia negotiating a separate

discussion during fieldwork. Clinics involving different professionals (e.g. a combination of GP, specialist nurse and third sector providers) were one way of ensuring that people with dementia and carers had space to express any concerns in private. While service users could contact some services by email, even this was felt to be problematic by some carers:

> [. . .] people do feel it's a betrayal. You don't want to put in writing, 'Dad's really aggressive now and he hit me last week.' They feel bad. But to actually say it, I can. . . I really feel for a lot of the relatives on that. It's really important that they have that two-room dynamic. (S303, service administrator)

Interventions to support relationships included specialist couples or family counselling. Carers who had experienced this type of counselling found it valuable in reducing stress and addressing their fears about the future:

> In the beginning I didn't feel that I could share how I was feeling. This couples counselling has actually really helped because we've been able to share stuff, our fears and our worries with somebody else there. That has helped quite a lot. I think that's taken away quite a bit of the stress. (C202, carer)

**Practical support.** This theme covers components which focus on enabling people with dementia and carers to live independently and safely for as long as possible (Table 6). While some elements of this theme focus on the needs of the person with dementia (such as support with activities of daily living or provision of assistive technology) or the carer (such as having a break), all of the components in this theme are relevant to both people with dementia and their families. Advocacy and safeguarding (protection from harm), for example, could include support for carers with decision making. While maintaining independence was identified as an important component of post-diagnostic support by several people with dementia in our study, feeling safe at home was also valued. A key aspect of this theme is balancing these two needs.

Finding acceptable levels of risk could be an area of conflict between people with dementia and carers, requiring a sensitive and thoughtful approach. This may involve environmental reviews and the introduction of assistive technology, as well as educating those involved with supporting a person with dementia:

> Don't–what's the word?–mollycoddle them [. . .] If you're doing the toast, get a toaster. They're still independent. Instead of using the gas [cooker], you can use a toaster. You've just got to change. . . We've got a toaster now, so I don't have to use the gas now. (P203, person with dementia)

> Just giving them [carers] a bit of education, and sometimes it's reassuring them that, yes, he is living at home with risks, but people are allowed to live at home with risks. It's just managing those risks in the best possible way, and reassurance for them. (S212, dementia practitioner)

Participants emphasised the need to provide practical help rather than purely signposting. While provision of information about financial support was valued by people with dementia and carers, who were frequently unaware of the social security and other benefits available, practical help with contacting services and completing forms was sometimes essential since forms could otherwise feel daunting and be abandoned, resulting in unclaimed benefits:

*Some of the things that we were sometimes struggling to get sorted, because you know, ringing up some of these organisations you're going to, you can't get through to some of them. The council. Knowing which page to go on or which form to fill in, where she'd got all that knowledge. So, [our named point of contact] facilitated, really, a lot of things that we wanted to put in place and made it flow a lot more easily. (C404, carer)*

Similar practical issues existed around other components. While cost was identified by carers as a potential barrier to having a break, it was not the only one; one carer described the burden of searching for a service that could provide an appropriate break for her husband with Lewy body dementia:

*You're ringing round homes wanting to know do they do respite? How long for? Who does? Who doesn't? That takes up a lot of time and effort [. . .] it would have been helpful if somebody had then said, "Right. Here's a list. This is who you get in touch with or these are the care agencies that are good. Use one of those." So you don't have to vet [check] them all yourself. They've already vetted them. (C608, carer)*

**Integrating support.** Given the multiple providers and sectors involved in delivering the components of post-diagnostic support described above, mechanisms are needed to ensure that services are well integrated and coordinated throughout the entire illness trajectory. This theme therefore relates to integration of post-diagnostic support and comprises three components (Table 7).

A key issue relating to this theme is the distinction between a named point of contact and a care coordinator. Professionals viewed these as very different roles, with many participants arguing that care coordination was only relevant to people with dementia and carers with complex needs where multiple agencies were involved. The need for care coordination in these circumstances was clearly expressed by one carer:

*When he's in hospital and you're trying to work out, like we need help now [with] managing him coming home. Does he need any. . .? We've had the occupational therapist round and they're saying he potentially might need another pole on the stairs to help him go upstairs, and some better toilet seat. But also, because of his pressure sore and things like that, will he need a mattress? Duh duh duh duh duh. All these sort of things. Who provides them? (C606, carer)*

While the precise role of a named point of contact varied in different services, at a minimum they could be contacted for information and advice at any point on the illness trajectory. In some services, the named point of contact had a more proactive role, such as providing regular reviews. Some services provided a hub or team, rather than a single named individual since this was thought to be more sustainable. People with dementia and carers also saw benefit in this approach:

*You have somebody who is actually on your side and cares about you, hopefully, and knows them, like a GP would. However, the problem with that is when they go on holiday, if they are on training and all this sort of thing. So, a small team might be better if it was just small though. (C002, carer)*

In view of their complementary role, access to both a named point of contact and care coordinator could be needed at different times. Some services ran a tiered system, whereby most

people with dementia were allocated to a named point of contact (unregistered, not professionally qualified staff) but could easily be transferred to a registered professional, typically a nurse, for more intensive support. This enabled a seamless transition and avoided the need for referrals which could create delays in accessing support. Specialist dementia professionals embedded in primary care typically had a more blended role and adjusted the level of support as and when needed. People with dementia and carers undergo transitions during the illness trajectory; while some are generally relatively minor and straightforward (e.g. the transition from a specialist memory assessment services to primary care; accessing support services for the first time), others are more challenging and could require intensive support (e.g. hospital avoidance, admission and discharge; move to a care home). Relatively simple mechanisms could be used to facilitate transitions, for example, some memory assessment services automatically referred people with dementia and carers to social prescribers (unless they did not consent to this). These arrangements, however, were not consistent, leaving some people with dementia and carers feeling abandoned by services and unsure how to access support:

> *The consultant said he didn't want to see us again. He said, "I'll refer you to the memory clinic." We went to the memory clinic once and they said, "Well, there's nothing we can do for FTD [frontotemporal dementia]. There's no medication, there's nothing, so we won't want to see you again." [. . .] So, we did feel a bit abandoned, as if we'd just been dropped, when they said they were just discharging us from everything. (C402, carer)*

Supporting the transition to a new service could include help with transport, or by a known worker accompanying the person to the service for the initial visits. While not all services offered this support, it was clearly valued by those who had received it, and could enabled people with dementia to access services at an earlier stage than might otherwise have happened. The first quote below is from a person living with young onset dementia, the second from his wife illustrating how the worker helped him to overcome his hesitancy:

> *'You know what it is? With new things I can talk myself out of it.' (P501, person with dementia)*

> *[Dementia navigator] has been really good for [my husband] because she's encouraged him, where something coming from me might not go down as well as somebody external coming in saying, "Look, tell you what, I'll pick you up at 10 o'clock and we'll go together," so that's been really, really helpful. (C501, carer)*

Supporting (or avoiding) other transitions required more intensive input. This was achieved in some services by specialist teams who could provide intensive support, typically to avoid hospital admission. Another service had access to short-term, 'flexi-beds' within a nursing home; these could provide support but with the explicit goal of enabling people to return to their own home when safe to do so:

> *[S202, enhanced support practitioner] gave an example of a success story of a woman who would have been sectioned [compulsorily admitted to a mental health hospital] because of threats of violence towards her daughter, but S202 was able to avoid admission by getting her into a flexi-bed for several months while they adjusted her medication and got her stable. When she went home, S202 said the daughter was delighted and said she had got her old mother back. (Fieldnotes of home visit, site 2)*

## Discussion

This paper is the first to our knowledge to use extensive qualitative research to identify the components of good post-diagnostic support for people living with dementia and their carers. This work supplements existing national evidence-based guidance [25] for England and Wales and frameworks for post-diagnostic support [11,35,36] by specifying the full range of components required to enable people to live as well as possible with dementia from the point of diagnosis through to end of life. These components have the potential to act as a benchmark and to drive much needed improvements in the consistency and breadth of post-diagnostic support in dementia. Comparison of our components to published syntheses of outcomes valued by people with dementia and carers [37–39] indicates that all of the outcomes listed in these syntheses can be mapped onto our five main themes, suggesting that it provides a comprehensive framework. Further, we believe that by emphasising the overlapping needs of people with dementia and carers (Fig 1), the framework highlights the broad range of carer needs. This could potentially facilitate more holistic carer assessment which would ensure that, for example, physical health is explored as well as specific support needs relating to the caring role.

We have not made recommendations relating to who or how the components should be delivered; instead we believe local solutions are needed which take existing service configurations and resources into account. Our findings highlight the importance of mechanisms for: i) reviewing which components are relevant for individuals at a specific point in time; and ii) integrating care and support within and between sectors. One potential mechanism for holistic, proactive review is the annual dementia review, which forms part of the Quality and Outcomes Framework for general practice in England, Wales and Northern Ireland [40]. Currently, however, there is no consistent template for these reviews and concerns have been expressed over the limited proportion of patients receiving reviews and their quality [41,42]. The five themes identified in our work offer a structured approach to the review and suggest that involving social care and third sector professionals could help identify non-medical needs. This could also promote greater integration of services, which remains challenging, particularly for people with dementia and carers who may need support from multiple services [18,43]. Improved collaboration is a policy priority in England, with the introduction of Integrated Care Systems which aim to address systemic barriers to joined up services [44]. We identified three components relating to integration, all of which are covered in national dementia guidance for England, although implementation varies [45]. We have reported on strategies to facilitate integration elsewhere [19]. These include the development of care pathways, defined as 'organisation of care processes for a well-defined group of patients during a well-defined period' [46]. There are documented examples of dementia care pathways [47–49], but implementation challenges exist, not least the linear management approach implied within a care pathway framework which presents a major challenge for a complex illness like dementia [50,51]. In contrast, our components potentially offer a different approach, with the aim being to develop a 'constellation' of components to meet individual needs at a particular moment in time.

Research has shown that post-diagnostic care and support services in England and Wales remain inadequate and inequitable [7,10]. This situation is echoed in other parts of Europe, with limited post-diagnostic dementia support and care services available, especially for people in the more advanced stages of dementia [8]. Worryingly, these data were collected prior to the COVID-19 pandemic which has disproportionately affected people living with dementia in terms of mortality [52–54], cognition, functional abilities and neuropsychiatric symptoms and quality of life [14,15]. Negative effects have been reported for carers in terms of stress, psychological and emotional wellbeing [55–57]. The disruption to services and negative

consequences for people with dementia and carers highlight the need for more robust and flexible approaches to post-diagnostic support. Particular attention to components within the theme of understanding and managing dementia, may help to improve resilience and ability to self-manage where support is disrupted.

## Strengths and limitations

A key strength of this work is that the components were identified through extensive qualitative research with a diverse range of providers [5,19], demonstrating that they can be delivered in practice. We also included a diverse range of service users in terms of age, diagnosis, time since diagnosis and living arrangements. We aimed to make the analysis as trustworthy as possible by interrogating the data drawing on our different backgrounds and experiences, before collectively discussing and coding the data. The components were further developed and validated through an iterative process with external experts, including people with dementia and carers, and comparison with guidelines [25] and frameworks [35,36]. Further strengths are the emphasis on the needs of both people with dementia and carers, and the shift away from linking interventions to specific points on the illness trajectory [11].

Limitations include some difficulties in recruiting a range of stakeholders in all case study sites, and it is probable that there was some selection bias since we relied on service managers to identify potential interviewees. Our focus was on services with strong links to primary care; while we included a range of professionals and services based in social and community care, there may be additional components specific to these sectors that we have not identified. Furthermore, due to the focus on components currently delivered in practice, there may be other desirable components that were not identified; however, we believe that our validation process would have uncovered these. Data were collected from England and Wales; the direct relevance of our components to people with dementia and carers globally is therefore unproven but this work provides a template for further study.

## Conclusions

The 2022 World Alzheimer Report will focus on post-diagnostic care, with the aim of addressing the international challenge of providing cost-effective and comprehensive post-diagnostic support for dementia. Our components potentially offer a framework for holistic, integrated and continuous care throughout the illness trajectory. This will require a focus on systems rather than the development, implementation and evaluation of individual interventions in isolation, which has characterised recent dementia care research [1,25,58]. Further research is needed to explore the relevance of our components to other sectors, countries and care systems. We will use our data [5,19] to develop and test a primary care-based intervention which aims to: develop systems; build capacity and capability; and deliver holistic, tailored post-diagnostic support for people with dementia and their carers.

## Supporting information

**S1 File. Reporting checklists.** Standards for Reporting Qualitative Research (SRQR) and Consolidated Criteria for Reporting Qualitative Research (COREQ).
(PDF)

**S2 File. Topic guides.**
(DOCX)

## Acknowledgments

Administrative support was provided by Angela Mattison. We thank the PriDem stakeholder panel and all participants for their assistance and involvement in the study. The PriDem study team is led by Professor Dame Louise Robinson at Newcastle University (a.l.robinson@ncl.ac.uk) and also includes Alistair Burns (NHS England and NHS Improvement), Sophie Dimitriadis (International Longevity Centre), Derek King (London School of Economics), Martin Knapp (London School of Economics), Doug Lewins (PPI co-applicant), Greta Rait (University College London), Sue Tucker (PPI co-applicant), Kate Walters (University College London), Jane Wilcock (University College London) and Raphael Wittenberg (London School of Economics).

## Author Contributions

**Conceptualization:** Claire Bamford, Louise Robinson.

**Data curation:** Claire Bamford, Alison Wheatley, Greta Brunskill.

**Formal analysis:** Claire Bamford, Alison Wheatley, Greta Brunskill, Laura Booi.

**Funding acquisition:** Claire Bamford, Louise Allan, Sube Banerjee, Karen Harrison Dening, Jill Manthorpe, Louise Robinson.

**Investigation:** Claire Bamford, Alison Wheatley, Greta Brunskill.

**Methodology:** Claire Bamford, Alison Wheatley, Greta Brunskill, Louise Robinson.

**Project administration:** Claire Bamford, Alison Wheatley, Greta Brunskill, Louise Robinson.

**Supervision:** Claire Bamford, Louise Robinson.

**Validation:** Claire Bamford, Alison Wheatley, Greta Brunskill, Laura Booi, Louise Allan, Sube Banerjee, Karen Harrison Dening, Jill Manthorpe, Louise Robinson.

**Visualization:** Claire Bamford, Alison Wheatley, Laura Booi, Louise Robinson.

**Writing – original draft:** Claire Bamford, Alison Wheatley, Louise Robinson.

**Writing – review & editing:** Greta Brunskill, Laura Booi, Louise Allan, Sube Banerjee, Karen Harrison Dening, Jill Manthorpe.

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
