## [Decision Letter · Decision Letter 0]

3 Sep 2021

PONE-D-21-24598

Key components of post-diagnostic support for people with dementia and their carers: A qualitative study

PLOS ONE

Dear Dr. Bamford,

Thank you for submitting your manuscript to PLOS ONE. After careful consideration, we feel that it has merit but does not fully meet PLOS ONE’s publication criteria as it currently stands. Therefore, we invite you to submit a revised version of the manuscript that addresses the points raised during the review process.

We look forward to receiving your revised manuscript.

Kind regards,

Saeed Shahabi

Academic Editor

PLOS ONE

Journal Requirements:

2. Please include additional information regarding interview guide used in the study and ensure that you have provided sufficient details that others could replicate the analyses, and please include a copy as Supporting Information.

3. We noted in your submission details that a portion of your manuscript may have been presented or published elsewhere. 

[We have published two papers drawing on the same datasets. There is some overlap with the methods sections. The focus of the first paper was on professionals' views of a task-shifted or task-shared approach. The second paper explored barriers and strategies to the delivery of post-diagnostic support. No quotations in the present paper have previously been published and this paper contains a greater focus on the views of people with dementia and carers than the earlier papers. There is some overlap between the barriers to post diagnostic support and the components; however, the barriers paper focuses more on how services are delivered, whereas this paper explores what components are being delivered. We believe this paper is distinctly different to the previous publications and adds to the existing literature.] 

Please clarify whether this publication was peer-reviewed and formally published. If this work was previously peer-reviewed and published, in the cover letter please provide the reason that this work does not constitute dual publication and should be included in the current manuscript.

4. We note that you have stated that you will provide repository information for your data at acceptance. Should your manuscript be accepted for publication, we will hold it until you provide the relevant accession numbers or DOIs necessary to access your data. If you wish to make changes to your Data Availability statement, please describe these changes in your cover letter and we will update your Data Availability statement to reflect the information you provide

5. One of the noted authors is a group or consortium [PriDem study team]. In addition to naming the author group, please list the individual authors and affiliations within this group in the acknowledgments section of your manuscript. Please also indicate clearly a lead author for this group along with a contact email address.

Additional Editor Comments:

The reviews are in general favorable and suggest that, subject to major revisions, your paper could be suitable for publication. Please consider these suggestions, and I look forward to receiving your revision. Please organize your qualitative study in accordance with the consolidated criteria for reporting qualitative research (COREQ) and the standards for reporting qualitative research (SRQR), and then upload completed forms as supplementary files.

Reviewers' comments:

Reviewer's Responses to Questions

**Comments to the Author**

1. Is the manuscript technically sound, and do the data support the conclusions?

Reviewer #1: Yes

Reviewer #2: Yes

2. Has the statistical analysis been performed appropriately and rigorously? 

Reviewer #1: N/A

Reviewer #2: Yes

3. Have the authors made all data underlying the findings in their manuscript fully available?

Reviewer #1: Yes

Reviewer #2: Yes

4. Is the manuscript presented in an intelligible fashion and written in standard English?

Reviewer #1: Yes

Reviewer #2: Yes

5. Review Comments to the Author

Reviewer #1: This is a very interesting paper and in-depth qualitative approach trying to understand the different components of post-diagnostic dementia care in England and Wales. Overall, this paper offers important new insights. However, there are some minor and major issues which need addressing, including lack of participant data, inappropriate discussion of findings in the results section, lack of mentioning of social care in the Introduction and more broadly at the end of the paper, lack of discussion of key findings in the Discussion. Further details are below:

- Whilst the authors provide a very clear and in-depth background about the structure of care in the UK, they miss out social care and discussing the interface between health and social care - after all, social care is the key form of care that people with dementia should access after a diagnosis (if they are aware of services, get referred, have enough funds, find the services suitable, etc...). The authors should therefore clearly refer to social care in the introduction, as dementia care is not solely run by primary care.

- In the last paragraph of the intro, the authors should state what type of study they are conducting, as opposed to stating in the last sentence 'The aim of this paper...'.

- How were topic guides developed? Were people affected by dementia involved? What was the broader level of public involvement in the study? This is not mentioned and a lack of public involvement would be a clear limitation.

- Where are the background characteristics of participants? They are not stated or referred to in the results.

- There are no page numbers by the way.

- Under timely identification and management of current & future needs, the authors discuss the finding, which is not suitable for the results and sentences including references should be removed [lines 199-201].

- Under the following theme, it seems strange that there was no mention of other non-cognitive symptoms. Instead, there is only a focus on cognition and behavioural symptoms, as well as functional deficits at one point, but no mention of for example speech difficulties, or motor or vision problems, as occur in certain subtypes.

- Again, picking up on a previous comment, the authors should not refer to existing literature in the field, discuss their findings, or provide further background in the results section. please delete lines 301-303.

- is the named point of contact and the care coordinator not effectively the same, or at least should be? Otherwise again there would be too many names/people to remember to contact, whereas accessing and utilising post-diagnostic dementia care should be as simple and straightforward as possible.

- The authors do not discuss the findings in depth in the Discussion, but provide a broad brush stroke of stating that this is a new guidance for a holistic PDS model. Furthermore, there is no mentioning of the pandemic. Yes, data were collected prior to the pandemic, yet COVID-19 has flipped PDS on its head and made it even more difficult to access care. Thus, the authors should refer to evidence in the field of how the pandemic has affected social care in dementia and possible implications for the long-term.

Reviewer #2: This is a well written, interesting article on a very important topic. There are some comments I have made for the authors to consider – but they are minor and I recommend the editor publish this article once the comments have been addressed. On a personal note, it’s nice to review such a well put together paper after reviewing so many dire ones of late – so thank you!

There needs to be researcher reflexivity in this article. I realise the interviews and analysis were conducted a while ago, but this is, in my opinion, not something that can be glossed over in qualitative research. So, how do the researchers feel their own practices, judgements, belief systems affect the data collection and analysis? How do you position yourselves in the research regarding your own background, work experience and other appropriate experiences? All this would add trustworthiness to the article and give it more depth. You could add this to the discussion – not much necessarily, but it needs adding and making clear.

A literature review was very recently published on integrated dementia care titled ‘Integrated dementia care: A qualitative evidence synthesis of the experiences of people living with dementia, informal carers and healthcare professionals’. Seeing as it is so relevant to this article and its recent publication, it would make sense to include here how your study extends knowledge from that review and how they link together. However you decide to address this comment, I feel the above article does need citing in your paper.

I am really very glad that you interviewed people living with dementia and included some of their comments in your findings section. There is so much research out there on similar topics where the voice of people living with dementia is missing completely, so I appreciated this. I often get very irritated when authors make recommendations for improving/changing systems or approaches to dementia care when only informal carers (also important) and health professionals have been consulted……or worse still just health professionals or academics. You interviewed a huge number of participants from diverse backgrounds and professions, informal carers and people living with dementia – I would add this to the ‘strengths and limitations’ section in the discussion of your article, but feel free to ignore this point if you feel it’s a bit OTT.

I am not very good at spotting typos (sorry!) – so hopefully the other peer reviewer(s) will have picked any up. But as I said previously, this is very well written paper so I doubt there are many.

6. PLOS authors have the option to publish the peer review history of their article (what does this mean?). If published, this will include your full peer review and any attached files.

Reviewer #1: No

Reviewer #2: No

---

## [Author Response · Author response to Decision Letter 0]

18 Oct 2021

Please see separate file 'Response to Reviewers' which addresses all points raised by the reviewers.

---

## [Decision Letter · Decision Letter 1]

11 Nov 2021

Key components of post-diagnostic support for people with dementia and their carers: A qualitative study

PONE-D-21-24598R1

Dear Dr. Bamford,

We’re pleased to inform you that your manuscript has been judged scientifically suitable for publication and will be formally accepted for publication once it meets all outstanding technical requirements.

Kind regards,

Saeed Shahabi

Academic Editor

PLOS ONE

Additional Editor Comments (optional):

Thanks for your revision. Please organize your manuscript in accordance with the consolidated criteria for reporting qualitative research (COREQ) and the standards for reporting qualitative research (SRQR) to enhance the reporting quality. Also, please upload a copy of them as supplementary file.

Reviewers' comments:

Reviewer's Responses to Questions

**Comments to the Author**

1. If the authors have adequately addressed your comments raised in a previous round of review and you feel that this manuscript is now acceptable for publication, you may indicate that here to bypass the “Comments to the Author” section, enter your conflict of interest statement in the “Confidential to Editor” section, and submit your "Accept" recommendation.

Reviewer #1: All comments have been addressed

Reviewer #2: All comments have been addressed

2. Is the manuscript technically sound, and do the data support the conclusions?

Reviewer #1: Yes

Reviewer #2: Yes

3. Has the statistical analysis been performed appropriately and rigorously? 

Reviewer #1: N/A

Reviewer #2: Yes

4. Have the authors made all data underlying the findings in their manuscript fully available?

Reviewer #1: Yes

Reviewer #2: Yes

5. Is the manuscript presented in an intelligible fashion and written in standard English?

Reviewer #1: Yes

Reviewer #2: Yes

6. Review Comments to the Author

Reviewer #1: The authors seem to have addressed the previous reviewers' comments. There is nothing I can add as new reviewer in this 2nd round.

Reviewer #2: The authors have adequately addressed all of my comments and I recommend this revised article for publication without the need for any further revisions. It is a very well written, interesting and, above all, an important paper which includes the voices of people living with dementia (a crucial strength in my opinion which is so often overlooked).

Thank you.

7. PLOS authors have the option to publish the peer review history of their article (what does this mean?). If published, this will include your full peer review and any attached files.

Reviewer #1: No

Reviewer #2: No

---

## [Editor Report · Acceptance letter]

10 Dec 2021

PONE-D-21-24598R1 

Key components of post-diagnostic support for people with dementia and their carers: A qualitative study 

Dear Dr. Bamford:

I'm pleased to inform you that your manuscript has been deemed suitable for publication in PLOS ONE. Congratulations! Your manuscript is now with our production department. 

Kind regards, 

on behalf of

Dr. Saeed Shahabi 

Academic Editor

PLOS ONE